# Best Fit DNA-Based Cryptographic Keys: The Genetic Algorithm Approach

**DOI:** 10.3390/s22197332

**Published:** 2022-09-27

**Authors:** Pratyusa Mukherjee, Hitendra Garg, Chittaranjan Pradhan, Soumik Ghosh, Subrata Chowdhury, Gautam Srivastava

**Affiliations:** 1School of Computer Engineering, Kalinga Institute of Industrial Technology (KIIT) Deemed to be University, Bhubaneshwar 751024, India; 2Department of Computer Engineering and Applications, GLA University, Mathura 281406, India; 3Siemens Technology Services Private Limited, Mumbai 560100, India; 4Sri Venkateswara College of Engineering Technology, Chittoor 517127, India; 5Department of Mathematics and Computer Science, Brandon University, Brandon, MB R7A 6A9, Canada; 6Research Centre for Interneural Computing, China Medical University, Taichung 40402, Taiwan; 7Department of Computer Science and Math, Lebanese American University, Beirut 1102, Lebanon

**Keywords:** DNA cryptography, genetic algorithm, data encryption, best key, key generation

## Abstract

DNA (Deoxyribonucleic Acid) Cryptography has revolutionized information security by combining rigorous biological and mathematical concepts to encode original information in terms of a DNA sequence. Such schemes are crucially dependent on corresponding DNA-based cryptographic keys. However, owing to the redundancy or observable patterns, some of the keys are rendered weak as they are prone to intrusions. This paper proposes a Genetic Algorithm inspired method to strengthen weak keys obtained from Random DNA-based Key Generators instead of completely discarding them. Fitness functions and the application of genetic operators have been chosen and modified to suit DNA cryptography fundamentals in contrast to fitness functions for traditional cryptographic schemes. The crossover and mutation rates are reducing with each new population as more keys are passing fitness tests and need not be strengthened. Moreover, with the increasing size of the initial key population, the key space is getting highly exhaustive and less prone to Brute Force attacks. The paper demonstrates that out of an initial 25 × 25 population of DNA Keys, 14 keys are rendered weak. Complete results and calculations of how each weak key can be strengthened by generating 4 new populations are illustrated. The analysis of the proposed scheme for different initial populations shows that a maximum of 8 new populations has to be generated to strengthen all 500 weak keys of a 500 × 500 initial population.

## 1. Introduction

Security of information in both transit and storage is extremely crucial to ensure its confidentiality, integrity, availability, and privacy. DNA cryptography [1,2] is one of the latest favourable techniques that has encompassed traditional cryptographic schemes with myriads of advantages. It is grounded on DNA computing where information is enciphered in the form of a DNA nucleotide by combining it with either a symmetric key or an asymmetric key [3,4]. Therefore, it combines the fundamentals of biological as well as mathematical concepts. The entire security of DNA cryptography relies on biological techniques and hence has no computation involved and is immune to attacks [5,6]. Moreover, information is represented in the form of four nitrogenous bases A, T, C and G thus the exponential power is four. Whereas in traditional cryptography, mostly binary values 0 and 1 are used giving an exponential power of two. Thus, each DNA encrypted bit becomes eight times stronger than its traditional counterpart. The constant endeavours to enhance the strength of cryptosystems further led to the implementation of the Genetic Algorithm (GA) in cryptography [7,8]. GA is an adaptive search algorithm which exploits the fundamentals of natural selection and genetics. It is used to solve problems with the help of evolutionary biological mechanisms like selection, crossover and mutation [9,10].

The Genetic Algorithm procedures begin with a random initial population which is composed of several individual chromosomes. In the majority of the cases, the chromosomes are binary i.e., either 0 or 1. Next, various operators are then iteratively applied to this initial population to create a better population. During selection [11], most fit chromosomes are chosen from the population to build the next generations of the population. The higher the fitness value, the more chances to be selected. The crossover [12] genetic operator joins two parent chromosomes to generate child chromosomes which are composed of chromosomes from each parent. The resultant child chromosomes are believed to possess the best characteristics of their parents and hence are considered to be more fit. Different types of crossovers are possible such as single-point crossover, two-point crossover, and uniform crossover. Mutation [13] is the genetic operator that assures genetic diversity among each generation. After crossover, mutation changes at least 1 bit in the chromosome to reflect the aftermaths of natural surroundings on the genetic procedures. This operation replaces the fewer fit chromosomes with the more fit ones. Like this, several generations of the population are created. Ultimately the best solution is selected from the updated population based on their probability or fitness function.

Fitness functions are extremely crucial as they enable effective exploration of the search space to get the best possible solution to the problem. The population is generated in keeping in mind that fit individuals are more likely to be replicated whereas unfit individuals are discarded. Cryptographic Key Selection is a type of selection problem where the key with the highest fitness and randomness ought to be selected. Therefore, this makes GA a reliable platform for key generation and selection. The application of Genetic Algorithms in traditional cryptography has been prevalent over the years.

The main contributions of this paper are highlighted as follows:Inculcate the benefits of Genetic Algorithms in DNA cryptography instead of Traditional Cryptography.Categorize the initial population of keys as strong or weak. The strong keys are used as it is for encryption. The weak keys instead of getting dropped are strengthened by the Genetic Algorithm. This step reduces the key generation time by only applying the scheme to weak keys. It also reduces key wastage.Propose suitable fitness functions by checking the frequency and gap of occurrence of the four nitrogenous bases to convert the weak keys into their fitter counterparts. It also reduces key wastage and enhances their efficiency for effective DNA-based cryptographic schemes.

This paper first provides a brief introduction to DNA Cryptography and explains why DNA-based schemes are more secure. It also touches upon the basic concepts of Genetic Algorithms. Section 2 focuses on related work of existing research. The proposed methodology is illustrated in Section 3 which first shows the basic block diagram of the overall steps involved and their descriptions. All the results obtained have been categorically demonstrated in Section 4. Section 5 provides the security analysis of the strengthened cryptographic keys. The conclusions drawn and the future scope of work are presented in Section 6.

## 2. Related Work

Soni et al. [14] proposed a simple genetic algorithm-based symmetric key generation scheme. Selection, Crossover and Mutation, the basic genetic operators are applied to the keys obtained from a random key generator to strengthen them. A similar GA-based cryptographic scheme was suggested by Singh et al. [15] to transfer secret information safely and securely. Their proposed scheme reads two consecutive bytes of the output of a binary random number generator and applies the common genetic operators on them to derive a safer key.

Mishra et al. [16] suggested that GA can be an excellent deciding factor to choose keys for public key cryptography after the keys are categorized on basis of their fitness.

Jhingran et al. [17] showcased how inculcating Genetic Algorithms into the RSA encryption scheme enhanced its immunity to attacks from intruders. Malhotra et al. [18] offered a genetic approach to generate a symmetric key for the IDEA algorithm to mitigate the occurrence of any weak keys in the process. They first categorized the possible weak keys that might be generated and then suggested techniques to make them use my performing crossover and mutation operations. Jain et al. [19] suggested a GA-based improvisation for OTP Key Generator. They emphasized the two main characteristics of speed and randomness both of which were improved by applying the genetic operators on the initial pad obtained from the linear congruential generator.

Chunka et al. [20] proposed an efficient mechanism to establish an initial secret key from the entire population based on Roulette Wheel Selection and two fitness functions negotiated between the sender and receiver beforehand. Suitable crossover and mutation operations were applied to get the improved population. Finally, the fittest one was designated to be the initial key which is then utilized to generate dynamic keys to perform the actual encryption of original data. The implication of GA to enhance the security of cryptosystems also garnered the interest of Nazeer et al. [21]. They first generated a key through a random number generator and then applied genetic operations to it. Next, they diffused the plaintext also by similar genetic operators. Finally, logical operations were performed amidst the diffused plaintext and the key to conduct the encipherment.

Kalsi et al. [22] proposed an interesting key generation scheme using GA. They generated their initial key population and applied the common genetic operators; however, they applied Run Test and Needleman- Wunsch Algorithm to check the randomness and degree of similarity to check the fitness of the newly generated population of keys. The most random and least similar key is chosen for encryption.

A combination of tree parity machines and Genetic Algorithms was utilized by Turčaník et al. [23] to generate the encryption keys at both the sender and receivers’ sites. They generated the initial population from the tree parity machine by manipulating the synaptic weights and then went on to apply the aforesaid genetic operations.

A novel key generation scheme for DNA cryptography was suggested by Vidhya et al. [24] using Genetic operators and the well acclaimed Diffie Hellman Key Exchange Protocol. The first DNA encoded the original plaintext from its ASCII values converted to their binary counterpart. Next, they generated two secret keys using the basic Diffie Hellman protocol onto which they applied several genetic algorithms to get more random and secure keys. One of the keys was used for single-point crossover and the other for bitwise mutation. Finally, they calculated the DNA sequence of their ciphertext which is transmitted to the receiver.

Tahir et al. [25] proffered a new model CryptoGA which is a genetic algorithm-based cryptosystem to cope with data security and privacy issues in cloud computing. Abduljabbar et al. [26] presented an encryption approach based on genetic operators. In this scheme, they divided the original message characters into pairs, and applied crossover onto them, followed by mutation to achieve the encrypted text.

Salamudeen et al. [27] proposed an enhancement to Audio Cryptosystems by applying GA. Each initial audio sample is genetically engineered by applying several operators to yield the final cipher audio. Garg et al. [28] suggested a genetic algorithm and DNA cryptography-based encryption scheme for Fog Networks. Hussein and Ayoob [29] came up with a secure key generation scheme to enhance Vigenere Cipher using the concepts of GA.

After scrutinizing the related work in Table 1, it can be observed that:The majority of the existing schemes are based on traditional binary keys and much less emphasis has been made on DNA-based keys.Most existing algorithms discussed are applying their proposed methodology to the initial key population which makes the key generation process lengthy and difficult.Based on the suitability of their proposal, each algorithm has defined its fitness test and selection, crossover, and mutation are the predominant genetic operators used.

Thus, the prime motivation of this paper is to introduce a scheme for enhanced security of DNA-based cryptographic keys as DNA cryptography is an upcoming field and traditional binary keys cannot be used there,
To choose the appropriate fitness test to be used as four different nitrogenous bases are involved in DNA cryptosystems.To decide whether the methodology is to be applied to the initial key population or not. For this, the fitness test is applied, and keys are categorized as strong or weak. If found strong, they are directly used for encryption. Only the weak keys are acted upon and thus the number of keys to be acted upon is reduced and the time complexity will reduce.To reduce key wastage by strengthening the weak keys and removing visible patterns instead of completely discarding them.

## 3. Proposed Methodology

An essential criterion for a secure cryptographic key is that the length of the key must be greater than or equal to the size of the original plaintext. Moreover, to add to the security of the cryptosystem, the same keys should not be repeated in another encryption. Thus, an obvious choice to quickly generate a large number of probable keys is through a pseudo-random key generator. Such keys, while not completely random, are still random enough for cryptographic purposes. However, amongst the generated keys, some keys might turn out to be weak and are rendered useless for encryption as they are vulnerable to attacks from intruders. The sole purpose of this paper is to design an algorithm to strengthen such weak keys using the concepts of Genetic Algorithms and make them usable to continue with the encryption process. Figure 1 gives the basic flowchart of the steps of the proposed methodology after which the sub-steps are explained in detail.

### 3.1. Generating the Initial Population

Let *N* be the number of DNA strings and *M* be the length of each DNA string. The number of DNA strings must be considerably large to encrypt a large chunk of the original message. The length of the DNA string is crucial as it needs to be greater than or equal to the length of the plaintext. The first essential task is to choose suitable values of *N* and *M* to generate the *N* × *M* initial population through a Random Key Generator. There are three possibilities: *N* < *M*, *N* = *M*, or *N* > *M*.

### 3.2. Applying Fitness Tests

After the initial population is generated for a particular value of *N* and *M*, both strong and weak keys are identified. Two fitness tests: The frequency Test and the Gap Test are applied to the initial population to test the strength of the keys.

The Frequency Test is performed to check the randomness of the key. It checks the frequency or number of occurrences of each nitrogenous base A. T, C and G in each DNA string. In this paper, if their respective frequency is nearly 25% of the length of the key, the key is considered strong, otherwise, it is categorized as a weak key. Another test known as the Gap Test is performed to determine the interval between two successive occurrences of similar nucleotides. In this paper, up to three successive repetitions of A, T, C or G are allowed to consider the string as strong. More than three repetitions enable intruders to identify a probable pattern in the resultant encrypted ciphertext thus making them prone to attacks. Therefore, after applying the fitness tests, the keys are either categorized as strong or weak. If strong keys are obtained, they can be directly utilized in encryption procedures. If weak keys are obtained suitable genetic operators are applied to strengthen these keys

### 3.3. Defining Fitness Functions for Weak Keys

Two Fitness Functions λ_1_ and λ_2_ are defined based on the Frequency Test and the Gap Test, respectively. Next *λ* is calculated by summing the obtained values of λ_1_ and λ_2_. Finally, the Fitness function *F* is obtained. All steps are illustrated next.

To calculate λ_1_ let the total number of weak keys be *n*. The frequency or number of occurrences of A, T, C, and G are stored in four variables A, T, C, and G, respectively. The ideal value of frequency which is approximately 25% of the length of keys is stored in the variable *i*. Next, the concept of standard deviation is applied to find the deviation of obtained frequency from the ideal frequency for each of the four nucleotides and stored in σ_A_, σ_T_, σ_C_ and σ_G_. Finally, λ_1_ is calculated as the average of σ_A_, σ_T_, σ_C_ and σ_G_. Equations (1)–(5) give the necessary formulas to calculate λ_1._
(1)$$\sigma_{A}=\sqrt{\frac{{\left(i-a\right)}^{2}}{n}}$$
(2)$$\sigma_{T}=\sqrt{\frac{{\left(i-t\right)}^{2}}{n}}$$
(3)$$\sigma_{C}=\sqrt{\frac{{\left(i-c\right)}^{2}}{n}}$$
(4)$$\sigma_{G}=\sqrt{\frac{{\left(i-g\right)}^{2}}{n}}$$
λ_1_ = (σ_A_ + σ_T_ + σ_C_ + σ_G_)/4(5)

Let λ_2_ be a flag to show which DNA string has more than three repetitive occurrences for any of the four nucleotides. Each of the 14 weak keys is scrutinized. If such a scenario for any of the A, T, C or G is obtained, λ_2_ is made 1 else 0.

The value of λ is calculated by summing the individual values of λ_1_ and λ_2_ as given in Equation (6). Table 2 showcases the calculation of λ for each of the 14 weak keys of the initial 25 × 25 population.
(6)$$\lambda =\lambda_{1}+\lambda_{2}$$

The final Fitness Function *F* is calculated by the formula given in Equation (7). The keys faring well in the frequency test and gap test are considered more fit as compared to their other counterparts. Thus, a lower value of λ implies a better value of *F*.
(7)$$F=\frac{1}{1+e^{\lambda }}$$

### 3.4. Arranging in Decreasing Order of Fitness Function

After calculating fitness function *F*, the 14 weak keys are arranged in decreasing order of their fitness by simply comparing and sorting the values. Although the step is simple, it still holds a lot of importance, as the two fittest keys will be selected first and crossover will be applied to them to generate the newer population. The more keys there are, the stronger they will be to be used for encryption.

### 3.5. Perform Crossover Operation

After arranging the 14 weak keys obtained from the initial 25 × 25 population in decreasing order of their fitness, the next task is to choose the type of crossover operation as well as a suitable crossover point. The different possible crossovers are single-point crossover, two-point crossover, and uniform crossover. Single point crossover chooses one point on the parent strings and all data beyond that point is swapped between the two parents. In contrast, two-point crossover chooses two random points on the parent strings and all data between these points are exchanged between the two parents. Uniform contrast simply selects one bit in the parent string randomly and toggles it with the corresponding bit in another parent. Currently, this paper emphasizes single-point crossover and there can be three possibilities to choose the point. The preferred point can be towards the starting of the strings, at the midpoint or towards the end. To receive a somewhat balanced crossover, this paper chooses a crossover point in the middle. Finally, the child DNA strings are generated by applying single point crossover on each group of two-parent DNA Key strings arranged in decreasing order of fitness. If a parent string does not have a pair, it is left as is.

### 3.6. Perform Mutation Operation

This paper mainly utilizes the concept of mutation to try to distribute the frequency distribution to some further extent on the child strings obtained post crossover operation to make them comparatively fitter and useful for the encryption process. The steps involved are described next.

The frequency or number of operations for each of A, T, C, and G are recalculated for all child DNA strings for each child population obtained for the three crossover points. They are again stored in 4 variables A, T, C, and G, respectively. As already mentioned in Section 3.3, the ideal value for the frequency is *i*. The lowest and highest occurring nitrogenous base is identified by Min (*a*, *t*, *c*, *g*) and Max (*a*, *t*, *c*, *g*). This paper proposes to perform mutation on the nucleotide having the least number of occurrences with the nucleotide having the highest occurrence to somewhat distribute the frequency. If multiple nucleotides have the same Min value, only one can be chosen. If multiple nucleotides have the same Max values, choose the one that has more consecutive repetitions as this makes the key more eligible to pass the Gap Test in the future. Let *m* be the number of instances of the nitrogenous base to be muted. The formula for the same is depicted in Equation (8) and *m* instances of the highest occurring nucleotide are substituted with the least occurring nucleotide again making sure that the substitution does not lead to more than three consecutive repetitions.
(8)$$m=i-Min\left(a,\;t,\;c,\;g\right)$$

Next, substitute *M* instances of the least occurring nucleotide with the highest occurring one. However, while doing so, we must keep in mind that substitution does not lead to more than three successive occurrences consecutively.

### 3.7. Generate the New Population

Finally, the new population is generated with comparatively fitter children obtained after crossover and mutation operations. As per the fundamental concepts of GA, this new population is comparatively fit as compared to its preceding population and the chances of them passing the fitness tests are thus tentatively higher.

### 3.8. Reapply Fitness Test and Repeat the Entire Process

The fitness tests will be reapplied to this population to categorize them into strong or weak. The encryption process can be continued with strong keys. Fitness Function is redefined for the newly obtained set of weak keys and then they are rearranged in decreasing order of fitness. Similar Crossover and Mutation operations will be applied to the resultant child population.

Thus, the entire process is repeated until each weak key is strengthened and after applying the fitness test, all strings pass the test and no key is rendered weak.

## 4. Results and Calculations

This section explains the actual implementations for all the steps mentioned in the proposed methodology. All the necessary calculations are illustrated in different tables and necessary analysis has also been provided.

### 4.1. Generating the Initial Population

The proposed methodology has been implemented on a randomly generated key population. For ease of demonstration, the implementation is currently depicted for *N* = *M* = 25. Thus, we have 25 DNA strings each having 25 chromosomes. Figure 2a showcases a glimpse of the same.

### 4.2. Applying Fitness Tests

After generating the initial key population, the already mentioned fitness tests are applied to segregate the keys as either strong or weak. Here since *M* = 25, to get a uniform frequency distribution, the ideal value must be approximately 6. If the total number of occurrences of each of the four nucleotides A, T, C and G are not equal to the ideal value, or there are more than three consecutive repetitions of any of these or a pattern is observed among the keys, they are rendered weak. Figure 2b further illustrates the categorization of the initial 25 × 25 randomly generated keys into weak and strong. A total of 14 weak keys have been identified out of the 25 generated key strings. The 11 strong keys can be directly utilized for encryption procedures. The proposed scheme proceeds with these 14 weakly identified keys.

### 4.3. Defining Fitness Functions for Weak Keys

We have obtained 14 weak keys so *n* = 14. The ideal value of frequency which is approximately 25% of the length of keys is stored in the variable *i*. Since *M* = 25, this paper considers *i* = 6. Next σ_A_, σ_T_, σ_C_, σ_G_ and λ_1_ are calculated using Equations (1)–(5). Table 2 illustrates the entire calculation of λ_1_.

Each of the 14 weak keys is scrutinized to find a repetition of more than three consecutive occurrences of any of the four nucleotides. If such a scenario for any of the A, T, C or G is obtained, the value of λ_2_ is made 1 else 0. The process is demonstrated in Table 3.

The value of λ is calculated using Equation (6). Table 4 showcases the calculation of λ each of the 14 weak keys of the initial 25 × 25 population. The final value of *F* is calculated using Equation (7) and is also shown in Table 4.

### 4.4. Arranging in Decreasing Order of Fitness Function

Table 5 provides the Weak keys arranged in decreasing order of fitness by simply sorting and rearranging them. This will later enable us to apply genetic operators on the fittest keys first.

### 4.5. Perform Crossover Operation

Among the three possibilities mentioned in Section 3.5, this paper chooses the crossover point (CP) towards the middle of the strings. Thus *CP* = 12 is selected and crossover is performed. Figure 3 gives the newly generated child population after performing a crossover operation with *CP* = 12 on each pair of parents chosen sequentially from Table 5.

### 4.6. Perform Mutation Operation

The child populations obtained after crossover is next scrutinized for mutation. Table 6 illustrates this step for the child population shown in Figure 3. Like our previous calculations, *i* = 6. The Min (*a*, *t*, *c*, *g*) and Max (*a*, *t*, *c*, *g*) values are highlighted in bold for easy identification. If multiple nucleotides have the same Max values, choose the one having more consecutive repetitions as this we the key becomes more eligible to pass the Gap Test in future. Still, if all have identical occurrences, any nucleotide can be chosen randomly. *m* is calculated using Equation (8).

Next, *M* instances of the least occurring nucleotide are substituted with the highest occurring one as per Table 6. However, while doing so again it has to be made sure that substitution does not lead to many successive occurrences consecutively. The child string is traversed, and three or more consecutive occurrences of A, T, C, and G are noted. Within that portion of nucleotides, one occurrence is muted after verifying that this mutation does not lead to more than three continuous occurrences of the nucleotides in muted string. This verification is done by keeping a track of the nucleotide preceding and succeeding the concerned nucleotide A, T, C, and G before and after mutation Figure 4 represents the generation of muted string from the original child string. The particular instance which is being muted is also highlighted in Figure 4.

### 4.7. Generate the New Population

Finally, the first set of the new population is generated which comprises the muted strings as calculated in Figure 4. This new population is shown in Figure 5a.

### 4.8. Reapply Fitness Test and Repeat the Entire Process

The fitness tests are re-applied to the first new population to categorize them into strong or weak. Figure 5b represents the weak keys obtained in this new population.

After applying the proposed methodology to this new population of 14 weak keys, only 4 weak keys are encountered. Thus, the suggested technique could strengthen the majority of the keys and render them useful for the encryption process.

Next, the already mentioned steps are re-applied to these weak keys obtained from the new population. In this new population as per Figure 5b, 4 weak keys have been obtained. Table 7 shows the fitness function *F* calculation for the same. Based on Fitness value, they are arranged in decreasing order.

After this, a crossover is applied to this new population for again *CP* = 12. The corresponding child population for this first new population is shown in Figure 6a. A mutation is applied to this child population after calculating the value of *m* as shown in Table 8, the process is illustrated in Figure 6b. These muted strings form the second new population which is shown in Figure 6c and its corresponding strong and weak keys after applying the fitness tests are shown in Figure 6d. Only 2 weak keys are obtained. Therefore, the suggested technique reduces the number of weak keys significantly on each iteration.

Next, a crossover is applied to the second new population at *CP* = 12. The resultant child strings and muted strings are shown in Figure 7a. A mutation is applied to this child population after calculating the value of *m* as shown in Table 9 and the process is shown in Figure 7b.

After this, the previously mentioned steps are re-applied to these weak keys obtained from the third new population. In this new population, as per Figure 7c,d, only 1 weak key is encountered. Therefore, no crossover operation is performed. Directly mutation is applied as shown in Table 10 and Figure 8.

The fourth new population is generated from the muted strings as shown in Figure 9a. Its classification into strong and weak keys is shown in Figure 9b.

Therefore, after applying the fitness test to the fourth population, no weak keys are obtained. Hence, the proposed method could successfully strengthen all the weak keys obtained at each step from different populations.

## 5. Analysis of Proposed Methodology

### 5.1. Number of Crossover and Mutation

The crossover rate and mutation rate play a crucial role to decide the chances that the two strings exchange some of their parts. It is the number of times crossover and mutation operations occur. A high crossover and mutation rate means most of the child strings are made from crossover and mutation operations and are comparatively fitter. However, this adversely affects the runtime complexity of the system. Figure 10 shows the number of crossovers and mutation operations performed in each generation of the 25 × 25 initial populations. It shows that with each new population the number of genetic operations to be applied is reducing drastically which reduces the further complexity of the system.

### 5.2. Effect of Different Values of N and M on the Number of Weak Keys Achieved

This paper scrutinizes the effect of different values of *N* and *M* on the number of weak keys achieved for each of the cases. It is evident that as the values of *N* and *M* increase, the number of weak keys generated is drastically increasing which necessitates the fact that procedures should be implemented to strengthen these instead of discarding and rendering them useless. Table 11 gives a demonstration of the same.

### 5.3. The Comparison of Number of Populations Generated to Strengthen Weak Keys Using Proposed Algorithm

The comparison of the number of populations generated to strengthen weak keys using genetic operators is also represented in Table 12. It is evident that at most 8 new populations are generated to successfully convert weak keys into strong ones. Therefore, from an implementation point of view it is evident that for a 500 × 500 initial population of keys, all the 500 keys might fall out to be weak keys based on frequency and gap test. They can be strengthened by the proposed method by generating a maximum of 8 new populations by applying the crossover and mutation operations.

### 5.4. Immunity to Security Attacks

From a security point of view, the most prominent attack is the Brute Force attack. In Bruce Force attacks, the intruder conducts an exhaustive search over the entire key space to guess the probable key. Binary keys consider only 0/1 hence the exponential power is 2. In DNA keys, because of the usage of 4 nitrogenous bases and different values of *M* and *N*, the key space to be traversed can be calculated by 4^4^ × *M* × *N*. Thus, the exponential power is 4. Therefore, each DNA key component is 8 times stronger than its binary counterpart. Figure 11 illustrates that the size of the key space is increasing exponentially, thus making the keys less vulnerable to intrusions.

### 5.5. Complexity Analysis

The Genetic Algorithm’s time complexity is dependent on the selection operator, fitness function, as well as the type of genetic operators used. For single-point crossover, it is given by *O (g(nm + nm + n))* where *g* is the number of generations, *n* is the population size and *m* is the size of the individuals. Therefore, the complexity is on the order of *O(gnm))* by taking the leading term. The space complexity is always twice the population storage each time a new population is generated.

This time complexity is less compared to the time complexity of encryption schemes. The complexity of encryption schemes further depends on the components used for the same. Simple Permutation, Substitution, *XOR*, Split and Combine, as well as shifting operations take less time when compared to other advanced methodologies. Symmetric key schemes are faster than asymmetric key schemes as only one key must be generated and shared.

The overall complexity of encryption will depend on both the complexities of key generation and actual encryption. Strengthening the weak keys will be time taking than simply generating new key space each time and it will impact the overall encryption procedure. However, security will not be compromised if the proposed methodology is followed and the immunity to intrusions is enhanced at the cost of time and space complexity.

### 5.6. Practical Application of Proposed Scheme

DNA Cryptography is a promising area of Cryptography which is more secure than traditional cryptosystems because of the use of four nitrogenous bases as already discussed in Section 5.4. The proposed scheme gives a best-fit DNA key generation methodology and therefore will find avid application in any DNA cryptosystem. Any type of input, namely text-based or image-based, can be encoded in DNA string form by applying the appropriate DNA Encoding method. To further enhance the security, it can be treated by a suitable DNA key resulting in a double-layer encryption technique. Therefore, the enhanced key can be applied to any image encryption and confidential communication procedures.

## 6. Conclusions and Future Work

DNA keys are more immune to attacks compared to their traditional counterpart because of the utility of four nitrogenous bases in contrast to the binary 0s and 1s. However, if the key strings have a non-uniform distribution of A, T, G and C, or have more than three occurrences within the key, they become more predictable and vulnerable to intrusions. Therefore, such keys are categorized as weak keys. With increasing values of *M* (length of each DNA key string) and *N* (number of strings), it is evident that the number of weak keys generated is drastically high and sometimes, most of the keys are rendered weak. The proposed methodology strives to make such keys usable for encryption by strengthening them using the concept of genetic algorithms.

This paper provides a standard deviation-based Fitness Function calculation for the weak keys. These keys are arranged in decreasing order of their fitness, and the child population is generated by applying suitable genetic operators. It was seen that a maximum of 8 new populations had to be generated for a 500 × 500 initial population, where all 500 key strings failed the fitness test and were classified as weak initially. It is also evident that the crossover and mutation rates are reducing from 7 to 0 and 38 to 2, respectively with each new population as more keys are passing the fitness tests. From a security point of view, the proposed scheme is immune to Brute Force attacks as the key space to be searched is reaching to be exponentially large with different values of *M* and *N*. Even for the initial 25 × 25 population, the size of the key space is 160,000 tremendously high bases.

The future scope of work plans to investigate the effect of different crossover points over the initial populations. The clauses for the fitness tests can also be further modified to check the strength of the keys and make them less vulnerable to attack from intruders. Cheating [30], Spoofing [31], and Intrusion [32] Detection systems are garnering a lot of research interest and will be studied later on. Other domains such as Neural networks, Machine Learning, Blockchain Technology [33], and Game Theory based Authentication [34] and their applications in the field of Cryptology will be explored in future endeavours. Iris biometrics has recently gained a lot of popularity and efficiency in a variety of security applications. However, presentation attacks can target biometric systems. This attack is carried out by impersonating any biometric feature and appearing to be a real trait [35]. However, various spoofing attacks have been employed in recent years to compromise the security of a biometric system. A biometric liveness identification system uses a person’s specific biological characteristics to quickly and reliably identify them [36]. By distinguishing the characteristics of live fingerprints from those of phoney fingerprints, a liveness detection system can be developed to foil the various types of spoof assaults that could be launched. Over the course of the last few years, academics have proposed many methods that are based either on hardware or software [37,38,39]. Comparison of the hardened key with keys generated by other advanced algorithms, e.g., Range-gated laser image compression and encryption scheme based on bidirectional diffusion, novel image encryption algorithms based on least squares generative adversarial network random number generators can also be experimented on in further courses of action.

## Figures and Tables

**Figure 1 sensors-22-07332-f001:**
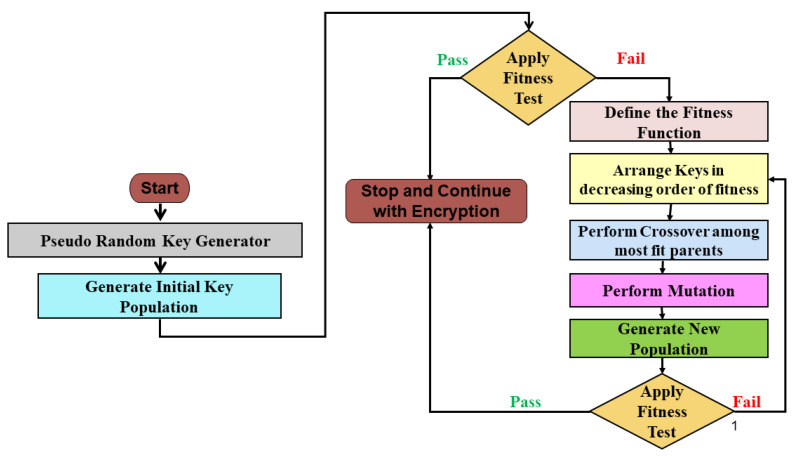
Block Diagram of Proposed Algorithm.

**Figure 2 sensors-22-07332-f002:**
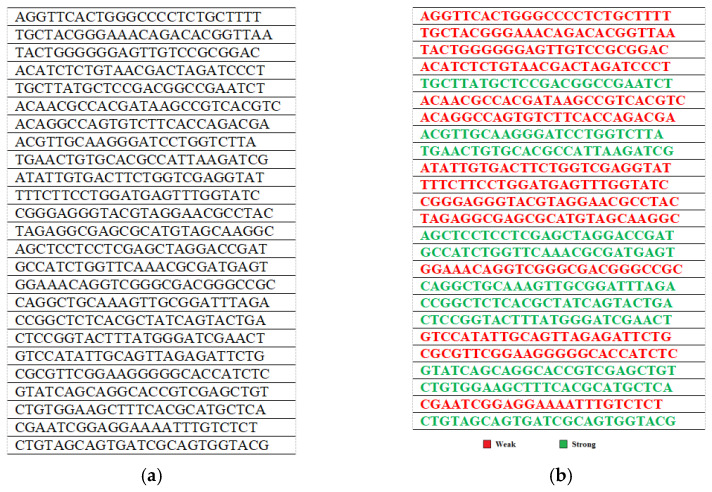
(**a**). Randomly Generated Initial 25 × 25. (**b**) Categorization of Keys among initial 25 × 25 Keys based on Fitness Tests.

**Figure 3 sensors-22-07332-f003:**
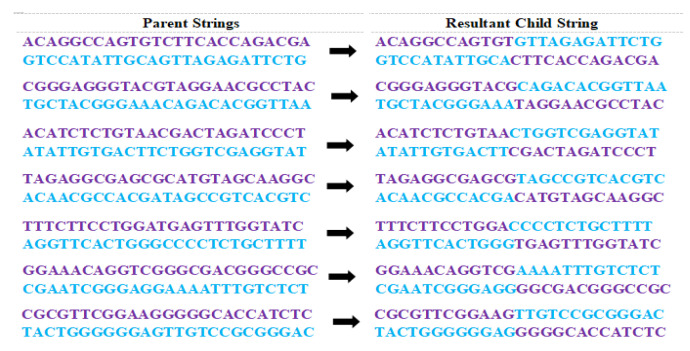
New Population after Crossover for Crossover Point (CP) = 12.

**Figure 4 sensors-22-07332-f004:**
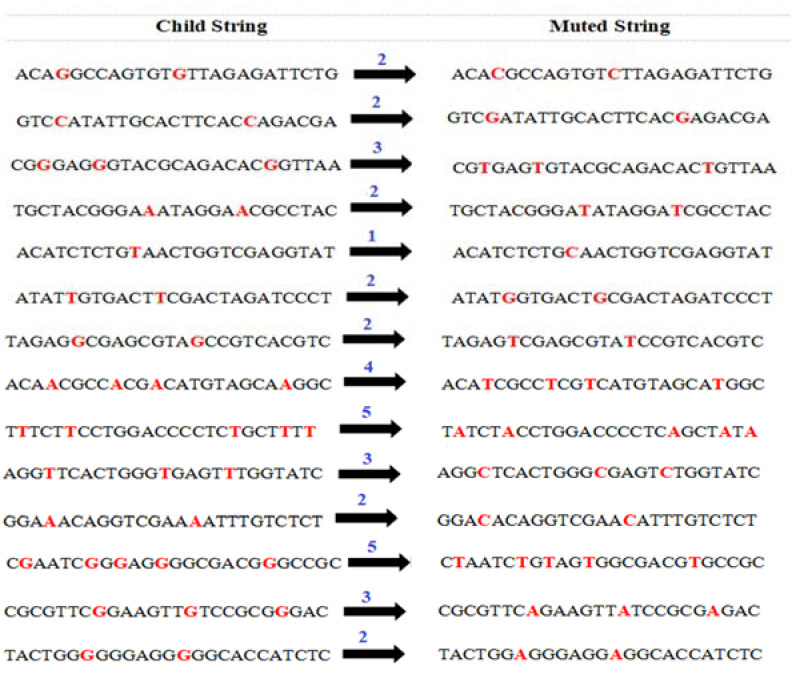
Mutation of Child Population.

**Figure 5 sensors-22-07332-f005:**
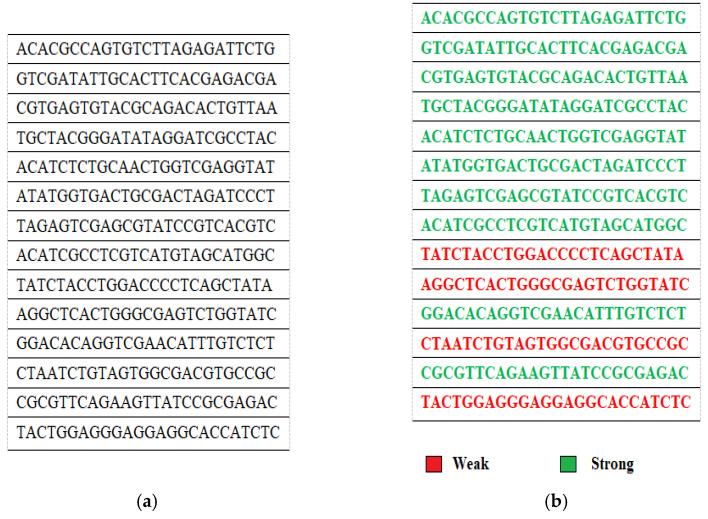
(**a**) First New Population after Crossover and Mutation. (**b**) Weak Keys in the First New Population based on Fitness Tests.

**Figure 6 sensors-22-07332-f006:**
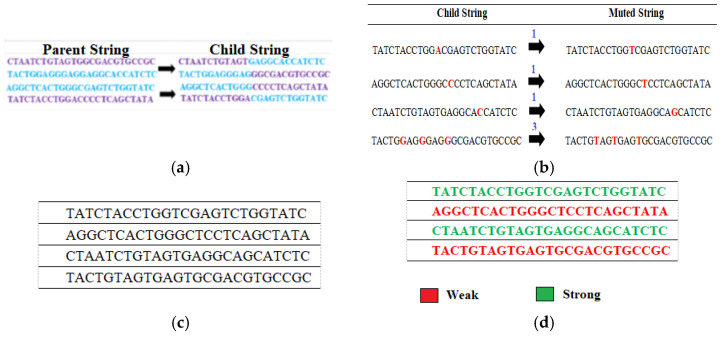
(**a**) Child Population from First New Population. (**b**) Mutation of Child Population from First New Population. (**c**) Second New Population after Crossover and Mutation. (**d**) Weak Keys in the Second New Population based on Fitness Tests.

**Figure 7 sensors-22-07332-f007:**
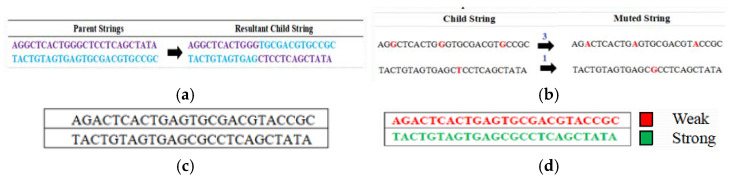
(**a**) Child Population from Second New Population (**b**) Mutation of Child Population from Second New Population. (**c**) Third New Population after Crossover and Mutation. (**d**) Weak Keys in the Third New Population based on Fitness Tests.

**Figure 8 sensors-22-07332-f008:**
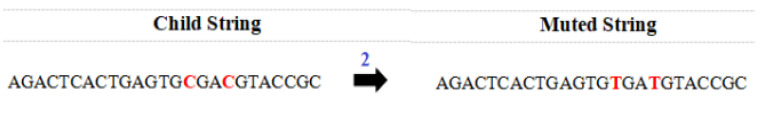
Mutation of Child Population from Third New Population.

**Figure 9 sensors-22-07332-f009:**
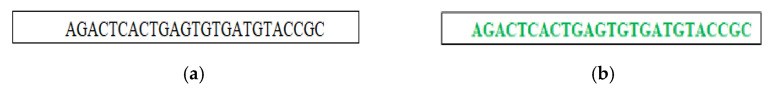
(**a**) Third New Population after Crossover and Mutation. (**b**) Weak Keys in the Third New Population based on Fitness Tests.

**Figure 10 sensors-22-07332-f010:**
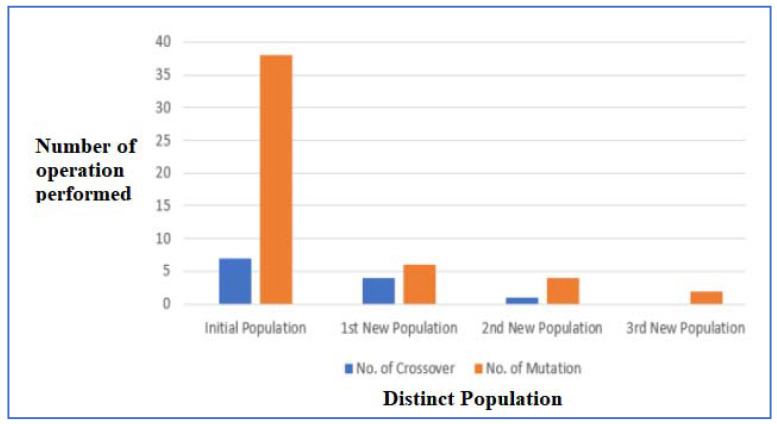
Comparison of Crossover and Mutation Rate in Different Generations.

**Figure 11 sensors-22-07332-f011:**
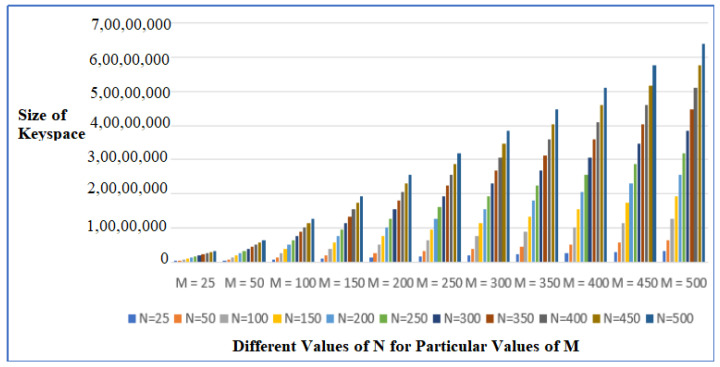
Size of Key Space to be searched for Brute Force Attacks for different values of *M* and *N*.

**Table 1 sensors-22-07332-t001:** Analysis of Related Work.

Author Name	Type of Cryptosystem	Genetic Operators Used	Fitness Test Applied	Whether GA-Applied on Complete Initial Key Population
Soni et al. (2012)	Traditional	Selection Crossover Mutation	Nil	Yes
Singh et al. (2013)	Traditional	Crossover	Nil	Yes
Mishra et al. (2013)	Traditional	Selection Crossover Mutation	Pearson’s Coefficient of auto-correlation	Yes
Jhingran et al. (2015)	Traditional	Selection Crossover Mutation	Nil	Yes
Malhotra et al. (2015)	Traditional	Selection Crossover Mutation	Comparing with parents	No
Jain et al. (2017)	Traditional	Selection Crossover Mutation	Frequency Test. Serial Test, Autocorrelation Test, Poker Test	Yes
Chunka et al. (2018)	Traditional	Selection Crossover Mutation	Frequency test, Block frequency, Runs test, Cumulative sums forward, Cumulative sums backward	Yes
Nazeer et al. (2018)	Traditional	Selection Crossover Mutation	Shannon Key Entropy	Yes
Kalsi et al. (2018)	DNA	Selection Crossover Mutation	Run Test and Needleman- Wunsch Algorithm	Yes
Turčaník et al. (2019)	Traditional	Selection Crossover Mutation	Frequency Test	Yes
Vidhya et al. (2020)	DNA	Selection Crossover Mutation	Shanon Key Entropy	Yes
Tahir et al. (2021)	Traditional	Selection Crossover Mutation	Shanon key Entropy	Yes
Abduljabbar et al. (2021)	Traditional	Selection Crossover Mutation	Nil	Yes
Salamudeen et al. (2021)	Audio	Bits fission Switching Mutation Fusion Deconditioning	Fission-Fusion Scheme	Yes
Garg et al. (2022)	DNA	Crossover Mutation	NA	Yes
Hussein et al. (2022)	Traditional	Crossover Mutation	Entropy Test	Yes

**Table 2 sensors-22-07332-t002:** Calculation of Fitness Function based on Frequency Test (λ**_1_**).

Weak Key	*a*	*t*	*c*	*g*	σ_A_	σ_T_	σ_C_	σ_G_	λ_1_
AGGTTCACTGGGCCCCTCTGCTTTT	2	9	8	6	1.069	0.802	0.535	0	0.6015
TGCTACGGGAAACAGACACGGTTAA	9	4	5	7	0.802	0.535	0.266	0.266	0.4673
TACTGGGGGGAGTTGTCCGCGGGAC	3	5	5	12	0.802	0.266	0.266	1.603	0.7342
ACATCTCTGTAACGACTAGATCCCT	7	7	8	3	0.266	0.266	0.535	0.802	0.4673
ACAACGCCACGATAGCCGTCACGTC	7	3	10	5	0.266	0.802	1.069	0.266	0.6008
ACAGGCCAGTGTCTTCACCAGACGA	7	4	8	6	0.266	0.535	0.535	0	0.3340
ATATTGTGACTTCTGGTCGAGGTAT	5	10	3	6	0.266	1.069	0.802	0	0.5343
TTTCTTCCTGGATGAGTTTGGTATC	3	12	4	6	0.802	1.603	0.535	0	0.7350
CGGGAGGGTACGTAGGAACGCCTAC	6	3	6	9	0	0.802	0	0.802	0.4010
TAGAGGCGAGCGCATGTAGCAAGGC	7	3	5	9	0.266	0.802	0.266	0.802	0.5340
GGAAACAGGTCGGGCGACGGGCCGC	5	1	7	12	0.266	1.336	0.266	1.603	0.8677
GTCCATATTGCAGTTAGAGATTCTG	6	9	4	6	0	0.802	0.535	0	0.3343
CGCGTTCGGAAGGGGGCACCATCTC	4	4	8	9	0.535	0.535	0.535	0.802	0.6018
CGAATCGGGAGGAAAATTTGTCTCT	7	7	4	7	0.266	0.266	0.535	0.266	0.3332

**Table 3 sensors-22-07332-t003:** Calculation of Fitness Function based on Gap Test (λ**_2_**).

Weak Key	λ_2_
AGGTTCACTGGGCCCCTCTGCTTTT	1
TGCTACGGGAAACAGACACGGTTAA	0
TACTGGGGGGAGTTGTCCGCGGGAC	1
ACATCTCTGTAACGACTAGATCCCT	0
ACAACGCCACGATAGCCGTCACGTC	0
ACAGGCCAGTGTCTTCACCAGACGA	0
ATATTGTGACTTCTGGTCGAGGTAT	0
TTTCTTCCTGGATGAGTTTGGTATC	0
CGGGAGGGTACGTAGGAACGCCTAC	0
TAGAGGCGAGCGCATGTAGCAAGGC	0
GGAAACAGGTCGGGCGACGGGCCGC	0
GTCCATATTGCAGTTAGAGATTCTG	0
CGCGTTCGGAAGGGGGCACCATCTC	1
CGAATCGGGAGGAAAATTTGTCTCT	1

**Table 4 sensors-22-07332-t004:** Calculation of Sum of Fitness Functions of Frequency and Gap Test (λ) and Final Fitness Function (***F***).

Weak Key	λ_1_	λ_2_	λ	*F*
AGGTTCACTGGGCCCCTCTGCTTTT	0.6015	1	1.6015	0.1868
TGCTACGGGAAACAGACACGGTTAA	0.4673	0	0.4673	0.3852
TACTGGGGGGAGTTGTCCGCGGGAC	0.7342	1	1.7342	0.1500
ACATCTCTGTAACGACTAGATCCCT	0.4673	0	0.4673	0.3852
ACAACGCCACGATAGCCGTCACGTC	0.6008	0	0.6008	0.3541
ACAGGCCAGTGTCTTCACCAGACGA	0.3340	0	0.3340	0.4181
ATATTGTGACTTCTGGTCGAGGTAT	0.5343	0	0.5343	0.3695
TTTCTTCCTGGATGAGTTTGGTATC	0.7350	0	0.7350	0.3241
CGGGAGGGTACGTAGGAACGCCTAC	0.4010	0	0.4010	0.4011
TAGAGGCGAGCGCATGTAGCAAGGC	0.5340	0	0.5340	0.3695
GGAAACAGGTCGGGCGACGGGCCGC	0.8677	0	0.8677	0.2958
GTCCATATTGCAGTTAGAGATTCTG	0.3343	0	0.3343	0.4171
CGCGTTCGGAAGGGGGCACCATCTC	0.6018	1	1.6018	0.1677
CGAATCGGGAGGAAAATTTGTCTCT	0.3332	1	1.3332	0.2113

**Table 5 sensors-22-07332-t005:** Weak Keys in decreasing order of Final Fitness Function (***F***).

Weak Key	*F*
ACAGGCCAGTGTCTTCACCAGACGA	0.4181
GTCCATATTGCAGTTAGAGATTCTG	0.4171
CGGGAGGGTACGTAGGAACGCCTAC	0.4011
TGCTACGGGAAACAGACACGGTTAA	0.3852
ACATCTCTGTAACGACTAGATCCCT	0.3852
ATATTGTGACTTCTGGTCGAGGTAT	0.3695
TAGAGGCGAGCGCATGTAGCAAGGC	0.3695
ACAACGCCACGATAGCCGTCACGTC	0.3541
TTTCTTCCTGGATGAGTTTGGTATC	0.3241
AGGTTCACTGGGCCCCTCTGCTTTT	0.1868
GGAAACAGGTCGGGCGACGGGCCGC	0.2958
CGAATCGGGAGGAAAATTTGTCTCT	0.2113
CGCGTTCGGAAGGGGGCACCATCTC	0.1677
TACTGGGGGGAGTTGTCCGCGGGAC	0.1500

**Table 6 sensors-22-07332-t006:** The number of instances to be muted (*m*) calculation for Child Population.

Child String	*a*	*t*	*c*	*g*	*i*	*m*
ACAGGCCAGTGTGTTAGAGATTCTG	6	7	**4**	**8**	6	2
GTCCATATTGCACTTCACCAGACGA	7	6	**8**	**4**	6	2
CGGGAGGGTACGCAGACACGGTTAA	7	**3**	5	**10**	6	3
TGCTACGGGAAATAGGAACGCCTAC	**8**	**4**	6	7	6	2
ACATCTCTGTAACTGGTCGAGGTAT	6	**8**	**5**	6	6	1
ATATTGTGACTTCGACTAGATCCCT	6	**9**	6	**4**	6	2
TAGAGGCGAGCGTAGCCGTCACGTC	5	**4**	7	**9**	6	2
ACAACGCCACGACATGTAGCAAGGC	**9**	**2**	8	6	6	4
TTTCTTCCTGGACCCCTCTGCTTTT	**1**	**12**	9	3	6	5
AGGTTCACTGGGTGAGTTTGGTATC	4	**9**	**3**	9	6	3
GGAAACAGGTCGAAAATTTGTCTCT	**8**	7	**4**	6	6	2
CGAATCGGGAGGGGCGACGGGCCGC	4	**1**	7	**13**	6	5
CGCGTTCGGAAGTTGTCCGCGGGAC	**3**	5	7	**10**	6	3
TACTGGGGGGAGGGGGCACCATCTC	**4**	4	6	**11**	6	2

**Table 7 sensors-22-07332-t007:** Calculation of Final Fitness Function (*F*) for First New Population.

Weak Key	*a*	*t*	*c*	*g*	σ_A_	σ_T_	σ_C_	σ_G_	λ_1_	λ_2_	λ	*F*
TATCTACCTGGACCCCTCAGCTATA	6	7	9	3	0	0.266	0.802	0.802	0.4675	1	1.4675	0.1873
AGGCTCACTGGGCGAGTCTGGTATC	4	6	6	9	0.535	0	0	0.802	0.3342	0	0.3342	0.4172
CTAATCTGTAGTGGCGACGTGCCGC	4	6	7	8	0.535	0	0.266	0.535	0.3340	0	0.3340	0.4172
TACTGGAGGGAGGAGGCACCATCTC	6	4	6	9	0	0.535	0	0.802	0.3342	0	0.3342	0.4172

**Table 8 sensors-22-07332-t008:** The number of instances to be muted (*m*) calculation in First New Population.

Child String	*a*	*t*	*c*	*g*	*i*	*m*
TATCTACCTGGACGAGTCTGGTATC	**5**	**8**	6	6	6	1
AGGCTCACTGGGCCCCTCAGCTATA	5	**5**	**9**	6	6	1
CTAATCTGTAGTGAGGCACCATCTC	6	**7**	**7**	**5**	6	1
TACTGGAGGGAGGGCGACGTGCCGC	4	**3**	6	**12**	6	3

**Table 9 sensors-22-07332-t009:** The number of instances to be muted (*m*) calculation for Second New Population.

Child String	*a*	*t*	*c*	*g*	*i*	*m*
AGGCTCACTGGGTGCGACGTGCCGC	**3**	4	8	**10**	6	3
TACTGTAGTGAGCTCCTCAGCTATA	6	**8**	6	**5**	6	1

**Table 10 sensors-22-07332-t010:** The number of instances to be muted (*m*) calculation for Third New Population.

Child String	*a*	*t*	*c*	*g*	*i*	*m*
AGACTCACTGAGTGCGACGTACCGC	6	**4**	**8**	**7**	6	2

**Table 11 sensors-22-07332-t011:** The number of Weak Keys Obtained for Different values of *N* and *M*.

	*M* = 25	*M* = 50	*M* = 100	*M* = 150	*M* = 200	*M* = 250	*M* = 300	*M* = 350	*M* = 400	*M* = 450	*M* = 500
** *N* ** **= 25**	14	22	24	24	24	25	25	25	25	25	25
** *N* ** **= 50**	22	38	44	49	49	49	50	50	50	50	50
** *N* ** **= 100**	34	87	94	97	98	98	99	99	100	100	100
** *N* ** **= 150**	60	89	95	112	139	146	147	148	149	150	150
** *N* ** **= 200**	79	98	126	157	164	179	198	198	199	200	200
** *N* ** **= 250**	89	99	135	168	173	191	240	248	249	250	250
** *N* ** **= 300**	107	115	142	196	248	289	291	298	299	299	300
** *N* ** **= 350**	137	141	156	198	249	324	335	340	350	350	350
** *N* ** **= 400**	148	159	175	180	237	329	367	384	400	400	400
** *N* ** **= 450**	158	173	192	226	290	316	384	437	450	450	450
** *N* ** **= 500**	173	213	246	287	314	384	453	488	497	500	500

**Table 12 sensors-22-07332-t012:** Number of Populations Generated to Strengthen the Weak Keys for Different values of *N* and *M*.

	*M* = 25	*M* = 50	*M* = 100	*M* = 150	*M* = 200	*M* = 250	*M* = 300	*M* = 350	*M* = 400	*M* = 450	*M* = 500
** *N* ** **= 25**	4	5	5	5	5	5	5	5	5	5	5
** *N* ** **= 50**	5	5	6	6	6	6	6	6	6	6	6
** *N* ** **= 100**	5	7	7	7	7	7	7	7	7	7	7
** *N* ** **= 150**	6	6	7	7	7	7	7	7	7	7	7
** *N* ** **= 200**	6	7	7	7	7	8	8	8	8	8	8
** *N* ** **= 250**	7	7	7	7	7	8	8	8	8	8	8
** *N* ** **= 300**	7	7	7	8	8	8	8	8	8	8	8
** *N* ** **= 350**	7	7	7	8	8	8	8	8	8	8	8
** *N* ** **= 400**	7	7	7	7	8	8	8	8	8	8	8
** *N* ** **= 450**	7	7	8	8	8	8	8	8	8	8	8
** *N* ** **= 500**	7	8	8	8	8	8	8	8	8	8	8

## Data Availability

There are no available data to be stated.

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
