# Peer review of "Best Fit DNA-Based Cryptographic Keys: The Genetic Algorithm Approach"

_sensors, 2022, doi:10.3390/s22197332_

Round 1

Reviewer 1 Report

1.The overall framework process of the article is very clear.

2.The variables used in some formulas in the article are not unified with those mentioned in the text,such as λ in formula 5. And some formulas are not displayed completely, such as formulas 6,7, etc. Please complete.

3.What is the difference between one-point crossing,two-point crossing and uniform crossing when performing crossover operation,please explain why.

4.Section 4.2 in the article mentioned 25 ests of keys in the fitness test. The key data in Figure 2b is insuffcient, please add.

5.I think the data in some tables can be combined into one table for dispaly, such as table 4, table 5.

6.When replacing the M least-occurring nucleotides with the most-occurring nucleotides, how do you choose the positions to replace ? How to ensure that substitutions do not result in consecutive identical nucleotides?

7.The proposed scheme improves the availability of the generated key. Will it take more time for the overall system when it is actually applied to the encryption scheme? Does it have an impact on the complexity of the system? Please analyze.

8.There are few experimental analysis, only security analysis is performed, please supplement the experimental analysis, such as time analysis efficiency analysis,complexity analysis,etc.

9.    Please verify the strength of the hardened key, compare the hardened key with keys generated by other advanced algorithms, e.g. Range-gated laser image compression and encryption scheme based on bidirectional diffusionA novel image encryption algorithm based on least squares generative adversarial network random number generator, etc.

    10.  I think that some practical applications can be appropriately added to the article, and the enhanced key can be applied to image encryption and confidential communication to reflect the actual value of the scheme, such as the application scenario Double image encryption algorithm based on neural network and chaos.

Author Response

Response to Reviewer 1

Comment 1: The overall framework process of the article is very clear.

Response: Thank you very much for acknowledgement.

Comment 2: The variables used in some formulas in the article are not unified with those mentioned in the text, such as λ in formula 5. And some formulas are not displayed completely, such as formulas 6,7, etc. Please complete.

Response: The variables mentioned in the article have been unified. Equations 1-7 have been accordingly rectified.

Comment 3. What is the difference between one-point crossing, two-point crossing and uniform crossing when performing crossover operation, please explain why.

Response: The difference has been explained in section 3.5 Page 7.

Comment 4. Section 4.2 in the article mentioned 25 ests of keys in the fitness test. The key data in Figure 2b is insufficient, please add.

Response: Figure 2a and 2b have been checked and updated.

Comment 5. I think the data in some tables can be combined into one table for display, such as table 4, table 5.

Response: Table 4 and Table 5 have been combined into Table 4 in Pg 11 and necessary revision in further table numbers have been made.

Comment 6. When replacing the M least-occurring nucleotides with the most-occurring nucleotides, how do you choose the positions to replace? How to ensure that substitutions do not result in consecutive identical nucleotides?

Response: The procedure has been explained in section 4.6 Page 14.

Comment 7. The proposed scheme improves the availability of the generated key. Will it take more time for the overall system when it is applied to the encryption scheme? Does it have an impact on the complexity of the system? Please analyze.

Response: The above points have been discussed and analyzed in section 5.5 Pg. No. 21.

Comment 8. There are few experimental analysis, only security analysis is performed, please supplement the experimental analysis, such as time analysis, efficiency analysis, complexity analysis, etc.

Response: The above points have been discussed and analyzed in section 5.5 Pg. No. 21.

Comment 9. Please verify the strength of the hardened key, compare the hardened key with keys generated by other advanced algorithms, e.g., Range-gated laser image compression and encryption scheme based on bidirectional diffusion A novel image encryption algorithm based on least squares generative adversarial network random number generator, etc.

Response: Brute Force attack is the most easy and prevalent attack on cryptographic as it is a category of Ciphertext Only Attack where the intruder has no other information apart from the ciphertext. Thus, as of now our research checks the immunity of the proposed scheme to survive a Brute Force Attack as in section 5.4. Pg. 20. Comparison of the hardened key with keys generated by other advanced algorithms, e.g., Range-gated laser image compression and encryption scheme based on bidirectional diffusion, A novel image encryption algorithm based on least squares generative adversarial network random number generator, etc. are currently beyond the scope of our research and will be tried upon in our future endeavors. Necessary changes in conclusion ad future work have been made in Section 6 Pg. 22.

Comment 10.  I think that some practical applications can be appropriately added to the article, and the enhanced key can be applied to image encryption and confidential communication to reflect the actual value of the scheme, such as the application scenario Double image encryption algorithm based on neural network and chaos.

Response: The practical application of the proposed scheme is discussed in section 5.6 Pg. 21.

Reviewer 2 Report

The manuscript presents the results of the research regarding the application of a Genetic Algorithm inspired method to strengthen the weak keys obtained from Random DNA-based Key Generators. The topic is exciting and actual. The manuscript is well structured; it contains all the necessary parts for this type of publication. To my mind, there are some disadvantages which should be corrected before the manuscript acceptance.

1. To my mind, the manuscript will look better if, after the section "Related Works", allocate the unsolved parts of the general problem and add the main contribution of the research.

2. I have questions to figure 1.  1) The block “Continue with Encryption” has two inputs. What is the output of this block? 2) What is the condition of this algorithm stopping? Please, presents this block chart correctly.

3. Please present the formulas correctly. Align numbering to the right.

4. Formulas 6 and 7. Are they correct?

5. Tables 2,… Title, Calculation of L1,… Please, present the correct title. Calculation of the fitness function (F) etc.

Author Response

Response to Reviewer 2

General Comment: The manuscript presents the results of the research regarding the application of a Genetic Algorithm inspired method to strengthen the weak keys obtained from Random DNA-based Key Generators. The topic is exciting and actual. The manuscript is well structured; it contains all the necessary parts for this type of publication. To my mind, there are some disadvantages which should be corrected before the manuscript acceptance.

Response: Thank you very much. The disadvantages have been addressed to the best of our capabilities in the updated manuscript.

Comment 1. To my mind, the manuscript will look better if, after the section "Related Works", allocate the unsolved parts of the general problem and add the main contribution of the research.

Response: Thank you for the suggestion. The necessary updation has been made after Table 1 in Pg. 5.

Comment 2. I have questions to figure 1.  1) The block “Continue with Encryption” has two inputs. What is the output of this block? 2) What is the condition of this algorithm stopping? Please, presents this block chart correctly.

Response: Figure 1 is redrawn. If the keys pass the fitness test, the algorithm stops and these keys can be directly used for encryption. If the keys fail the fitness test, they are further acted upon as suggested in the proposed methodology. It is discussed in Pg. No. 6.

Comment 3. Please present the formulas correctly. Align numbering to the right.

Response: Equations 1-7 have been checked and updated.

Comment 4. Formulas 6 and 7. Are they correct?

Response: Equations 6-7 have been checked and updated.

Comment 5. Tables 2,… Title, Calculation of L1,… Please, present the correct title. Calculation of the fitness function (F) etc.

Response: All table captions have been updated.

Round 2

Reviewer 1 Report

The problems in the article have been revised, and the reasons are clearly given. The experimental results also supplement the safety analysis and give the complexity analysis,and correspondingly indicate the practical application prospects of the scheme.  Experiments related to the practical application of the scheme can be supplemented in future work.

Reviewer 2 Report

Thanks, I have no other questions